# Dynamic networks observed in the nucleosome core particles couple the histone globular domains with DNA

Xiangyan Shi ⓘ [1,3 ✉], Chinmayi Prasanna ⓘ [2,3], Aghil Soman[2], Konstantin Pervushin[2] & Lars Nordenskiöld ⓘ [2 ✉]

The dynamics of eukaryotic nucleosomes are essential in gene activity and well regulated by various factors. Here, we elucidated the internal dynamics at multiple timescales for the human histones hH3 and hH4 in the Widom 601 nucleosome core particles (NCP), suggesting that four dynamic networks are formed by the residues exhibiting larger-scale µs-ms motions that extend from the NCP core to the histone tails and DNA. Furthermore, despite possessing highly conserved structural features, histones in the telomeric NCP exhibit enhanced µs-ms dynamics in the globular sites residing at the identified dynamic networks and in a neighboring region. In addition, higher mobility was observed for the N-terminal tails of hH3 and hH4 in the telomeric NCP. The results demonstrate the existence of dynamic networks in nucleosomes, through which the center of the core regions could interactively communicate with histone tails and DNA to potentially propagate epigenetic changes.

[1] School of Physical and Mathematical Sciences, Nanyang Technological University, 21 Nanyang Link, Singapore 637371, Singapore. [2] School of Biological Sciences, Nanyang Technological University, 60 Nanyang Drive, Singapore 637551, Singapore. [3]These authors contributed equally: Xiangyan Shi, Chinmayi Prasanna. ✉email: shi.xiangyan@ntu.edu.sg; larsnor@ntu.edu.sg

Genomic DNA in eukaryotic cells is organized into nucleosome core particles (NCP) that are further packed into chromatin fibers[1,2]. The NCP is formed by ~147 bp DNA wrapped around a histone octamer (HO) consisting of two H2A–H2B dimers and one (H3–H4)$_2$ tetramer[3]. Atomic structures have been solved for a number of nucleosomes, of which the majority are NCPs sharing highly identical structural features regardless of the differences in DNA sequences and histone variants. To precisely govern gene activities, eukaryotes evolve various mechanisms, such as post-translational modifications (PTMs) and recruitment of histone variants to promote or suppress DNA accessibility[4–6]. The PTM sites occur mainly on the histone N-terminal tails[7], however, novel PTMs recently discovered in the globular domains[4,5,8,9] suggest that the nucleosome core also mediates gene activities. H3 modifications, such as K56Ac, K64Me3, K79Ac, and K122Ac modulate transcription by altering the mobility and stability of DNA[5]. The underlying mechanisms of chromatin regulation by introducing PTMs are not fully understood. Modified residues in the histone globular domains often are inaccessible for direct interactions with chromatin factors, and the epigenetic impacts cannot be addressed by local structure changes alone. The HO in the nucleosomes form a well-compacted core, but also exhibits significant intrinsic plasticity that potentially drives DNA sliding and unwrapping. Recent Cryo-EM studies determined multiple conformations for a NCP at physiological conditions, suggesting that conformational rearrangement of the HO couple with DNA translocation[10,11]. The understanding of chromatin and nucleosome dynamics at atomic resolution and its functional relevance, although playing essential roles[12,13], is still limited. Furthermore, it was recently discovered that the activities of nucleosomes are modulated by many factors in an allosteric manner[14–17]. For example, the binding of the CENP-C to the CENP-A-containing nucleosomes alters the overall plasticity of HO, and remotely changes the sliding of the terminal DNA to achieve the activity regulation[15,17]. Another example is that the SNF2h remodeller binding to nucleosomes results in disorder in the distal sites to achieve allosteric control of DNA translocation[16], and various residues buried in the HO core also become more dynamic in this process[18]. Although an increasing number of nucleosome activities are discovered to be modulated in analogy with allosteric regulation, the mechanisms and pathways are unclear. It was hypothesized that the plasticity of the HO contributes and couples to the DNA translocation in many of those processes[10,11]. The NCP, which is the repeating unit of chromatin, assembles into the same structure with different DNA sequences, but exhibits variations in stability and DNA accessibility[19,20]. Telomeres reside at the end of chromosomes and the DNA consists of tandem repeats of guanine-rich sequences, such as TTAGGG in mammalian cells[21]. The critical roles of telomeres in protecting chromosomes urge the comprehensive elucidation of the structure and dynamics of telomeric nucleosomes to understand the molecular foundations of its unique biological behavior and function. A recently published study solved the XRD structure of a NCP reconstituted from 145 bp TTAGGG human telomeric sequence, and revealed a structure very similar to other NCPs with canonical DNA sequences, but an overall less stable complex in solution[22]. These unique dynamic properties seem to contribute to the biological activities of telomeric nucleosomes. However, the detailed dynamic information of telomeric nucleosomes remains unknown.

We and others have recently demonstrated that solid-state NMR (SSNMR) is uniquely suited for the characterization of nucleosomes dynamics and structure[23,24]. The study of the human histone H4 (hH4) dynamics in the Widom 601 NCP and nucleosome arrays elucidated that the sites exhibiting enhanced microsecond to millisecond (μs–ms) motions appears to overlap with functionally relevant regions. Further exploration of the dynamic properties for the rest of the nucleosome components is required to completely understand the contribution of dynamics to chromatin regulation. NMR is one of few premier techniques to probe the dynamics of biomolecules and is capable of quantifying atomic-resolution motions at second to picosecond timescales[25–29]. In addition, it provides higher spatial and temporal resolution in comparison with other common techniques, such as fluorescence spectroscopy and electron paramagnetic resonance, and does not require complicated labeling schemes. Molecular dynamics can provide rich information on atomic-resolution motions, however, is time-consuming for simulation of large complexes, such as nucleosomes when studying the slow motions. A recent methyl-TROSY solution-state NMR study demonstrated that the histone core in the NCP possesses significant dynamics[30]. In this study, we first elucidated the human histone H3 (hH3) dynamics and combined this with our previous results of hH4 to further explore the plasticity of the HO in the Widom 601 nucleosome at the atomic level. Furthermore, we probe the potential correlation with DNA dynamics by characterizing a nucleosome with telomeric DNA that lacks positioning information. We measured the internal dynamics of the hH3 in the Widom 601 NCP in the μs–ms and nanosecond to microsecond (ns–μs) timescales. Combined with our previous characterization of hH4 (ref. [23]), this suggests the presence of four distinct networks that comprise the hH3 and hH4 core residues exhibiting larger-scale collective motions in the μs–ms timescale, which propagate through the NCP core to the histone tails and DNA. These unique networks may transfer long-range changes induced by epigenetic modifications. Furthermore, the unique structural and dynamical features are explored for the N-terminal tails and the globular domains of the hH3 and hH4 in the human telomeric NCP, using a combination of SSNMR and liquid-state NMR. In comparison with the Widom 601 NCP, enhanced mobility was observed for residues in the identified dynamic networks and a spatially close motif in the telomeric NCP. This is the first study that reveals the existence of distinct dynamic networks in nucleosomes. The study indicates that the HO core can interactively communicate with histone tails and DNA through those unique networks, which may allow the regulation of gene activities by modifications of core residues in an allosteric manner. In addition, the revealed networks may serve as the starting point to discover the allosteric pathways of nucleosome modulation by many proteins.

## Results

**Nanosecond–microsecond dynamics of hH3 in Widom 601 NCP.** The complete SSNMR $^{13}$C/$^{15}$N assignments for the hH3 and hH4 core in the Widom 601 NCP have been obtained in our previous studies[23,31]. The elucidated NMR structures agree well with previous XRD studies. In addition, our study suggested that the internal dynamics of hH4 contributes to modulating DNA accessibility and, therefore, potentially gene activity. In order to obtain a more complete picture of the dynamics in the H3/H4 tetramer, we now also characterized the internal dynamics of the hH3. We first conducted three-dimensional (3D) dipole chemical shift correlation (DIPSHIFT)[32] SSNMR measurements for a NCP reconstituted from human histones and the 145 bp Widom 601 DNA to characterize the dynamics in the timescale of ns–μs in the hH3. Forty six residues in the hH3 globular domain are well resolved in the NCA dimension, of which the backbone $^{1}$H–$^{13}$Cα and $^{1}$H–$^{15}$N dipolar line shapes are extracted from the 3D experiments. Site-resolved dipolar order parameters, $S_{(CH)\alpha}$ and $S_{NH}$, derived from the line shapes are shown in Fig. 1, which report the motional amplitudes in the ns–μs timescale. Large-order parameters (>0.8) are determined for a majority of the residues. Residues in the hH3 LN (the loop between αN and α1)

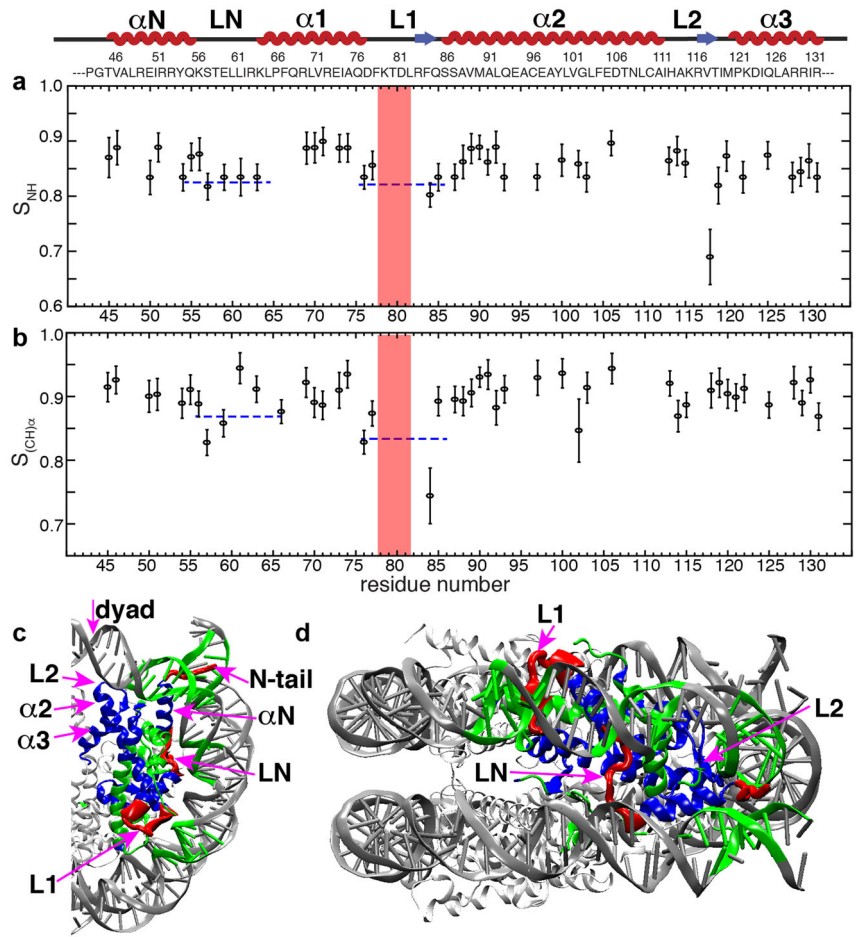

**Fig. 1 The hH3 in the Widom 601 NCP undergoes small-amplitude nanosecond–microsecond motions with slightly higher flexibility in the LN and L1.** **a, b** Dipolar order parameters $S_{NH}$ and $S_{(CH)\alpha}$ for hH3 in the Widom 601 NCP. The red box marks F78–D81 absent in the NCA spectrum. The dotted lines are given to guide visualization. **c, d** Regions of hH3 (one copy is shown) exhibiting relatively smaller dipolar order parameters are highlighted in red (PDB: 3LZ0). The N-tail and C-terminus residues absent in those dipolar-based experiments are also highlighted in red as they are highly dynamic. The rest of the hH3 residues having relatively larger dipolar order parameters are in blue. Sites of other histones and DNA that are ≤10 Å away from these regions (in red) are in green. The rest of the histones and DNA are shown as light and dark gray ribbons with smaller thickness, respectively. $^{1}$H-$^{15}$N and $^{1}$H-$^{13}$C dipolar line shape fitting used to extract $S_{NH}$ and $S_{(CH)\alpha}$ is provided in the Supplementary Fig. 2.

and L1 exhibit slightly smaller order parameters (compared to the rest of the globular domain) corresponding to larger-amplitude dynamics in the ns–μs timescale. The relatively looser packing of the coil structure determines the higher flexibility of LN and L1 in the ns–μs timescale. In comparison, the L2 has slightly larger-order parameters comparable to those of the helix regions, likely due to interaction with DNA through hydrogen bonds and salt bridges. It is noticed that the differences of the order parameters between the loop regions and the well-structured helices are rather small due to the compaction of the disc-like NCP core. Overall, the hH3 core in the Widom 601 NCP undergoes small-amplitude ns–μs motions and the absence of pronounced variations in these motions across the globular domain illustrates that NCP core is tightly packed, and provides overall structural stability and integrity for nucleosomes during genome activities.

**Microsecond–millisecond dynamics of hH3 in Widom 601 NCP.** Our previous study of nucleosome internal dynamics revealed that several regions in the hH4 globular domain exhibit larger-scale dynamics in the μs–ms time window, potentially contributing to chromatin regulation; however, the underlying mechanism of such regulation is unclear[23]. To further understand

the nucleosome dynamics in the H3/H4 tetramer in the nucleosome core and its possible functional relevance, we investigated the μs–ms motions for hH3 in the Widom 601 NCP. The cross-peak intensities in the dipolar-based heteronuclear correlation experiments, such as CANCO and NCA are determined by the efficiencies of multiple dipolar heteronuclear transfers, and can be used to qualitatively track the μs–ms dynamics that interferes with relaxation rates (T1ρ and T2*)[23,33]. Here, the CANCO peak heights are extracted for 76 residues of hH3 in the NCP (Fig. 2). The aa P43–R52 in the αN helix, I62–K67 in LN and α1 helix, and L1 exhibit lower CANCO intensities in comparison with the rest of the hH3, suggesting that they are dynamically less restricted in comparison with the rest core residues. L1 is the most dynamic region in the μs–ms timescale. In particular, the absence of CANCO correlations demonstrates the significant mobility of the F78–D81 region that resides at the DNA-free surface of the NCP and is absent from direct interaction with DNA. Recent studies showed that this stretch is involved in gene regulation and harbors several novel PTMs, including acetylation/methylation of K79 (ref. [34]) and phosphorylation of T80 (ref. [35]). The fact that this region does not interface directly with DNA leads to a possibility of remote signaling via dynamic networks. Indeed, this gene regulation mode is supported by the other dynamics properties

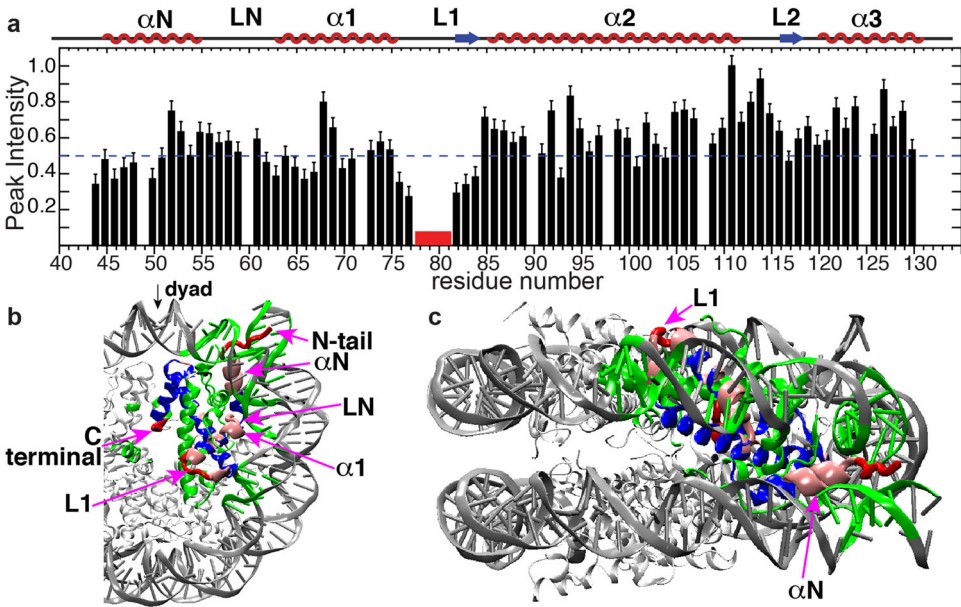

**Fig. 2 The hH3 in the Widom 601 NCP displays significant microsecond–millisecond dynamics in several regions in the globular domain. a** CANCO peak intensities of hH3 in the Widom 601 NCP. The red box marks F78–D81 which is absent in the spectrum. The peak intensities were normalized to the highest peak intensity in the dataset. The dashed line is drawn to guide visualization. **b**, **c** regions of H3 (one copy is shown) exhibiting negligible and relatively low peak intensities (<0.5) are in red and pink, respectively. DNA and other histone sites ≤10 Å away from these regions (in red or pink) are in green. The rest of the hH3 residues are displayed in blue. The rest histones and DNA are shown as light and dark gray ribbons with smaller thickness, respectively.

observed in the Widom 601 NCP and in the telomeric NCP, as discussed in the following. Relatively, larger-scale μs–ms dynamics is also observed in the majority of hH3 αN, including P43–R52 (Fig. 2). A previous study found altered nucleosome DNA sliding and unwrapping in H3 with site mutations in the αN and N-terminal tail[36]. The hH3 αN is located near the DNA entry–exit and is sandwiched between DNA, hH4 L1, and the hH2A C-terminus. A mutagenesis study suggested that the hH2A C-terminal loop participates in regulating nucleosome mobility[37]. In addition, we observed higher flexibility for the hH4 L1 (ref. [23]). Consequently, these dynamic regions seem to overlap with the histone residues linked to the functional response of modifications and mutations. Another hH3 motif showing relatively enhanced μs–ms dynamics is the stretch I62–L67 located at LN and α1 (Fig. 2b). Overall, three hH3 regions, including the αN, L1, partial LN and the adjacent residues in α1 exhibit relatively larger-scale μs–ms dynamics compared to the rest of the histone H3.

**Dynamic networks formed by the hH3 and hH4 in the nucleosome**. The previous study of hH4 in the NCP revealed that regions exhibiting higher dynamics in the μs–ms timescales harbor many residues significantly contributing to nucleosome biological activities[23]. Here, we summarize the dynamic regions of hH3 and hH4 in the Widom 601 NCP, as presented in Fig. 3. Remarkably, the hH3 and hH4 domains that display significantly more pronounced dynamics compared to the rest of the H3/H4 tetramer can be clustered together to form four dynamic networks: (1) the hH3 N-terminal tail, part of αN and H4 L1 (Fig. 3c); (2) hH3 I62–F67 in α1 and LN, the H4 N-terminal tail and H4 LN (Fig. 3d); (3) H3 L1 and part of the H4 N-terminal tail (Fig. 3e); and (4) the H4 C-terminal residues L97–G102 and part of H4 α3 (Fig. 3b). The first three dynamic networks directly interact with DNA and their enhanced mobility likely couple to the flexibility of DNA. The fourth network is buried in the NCP core and likely extends to the H2A C-terminus at the DNA-free surface of the NCP contacting the first network. Given that many

of these residues in these dynamic networks were previously found to be functionally relevant as discussed above, we hypothesize that these networks may couple epigenetic modifications in the nucleosome core with DNA, thus altering its mobility and accessibility. Furthermore, we hypothesize that differences in DNA sequences may reversibly induce changes in the NCP core via the same dynamic networks. To confirm this, we conducted SSNMR on a telomeric NCP reconstituted from a 145 bp telomeric DNA comprised of TTAGGG sequence repeats to probe the unique structural and dynamic features of hH3 and hH4.

**Structure and dynamics of hH3 and hH4 in human telomeric NCP**. The recent XRD structure of the telomeric nucleosome showed that the histones in the NCP reconstituted from the 145 bp human telomeric DNA fold into the same structures, as in other canonical NCPs[14]. Here, we first characterized the hH3 and hH4 structures in the precipitated telomeric NCP at more physiologically relevant conditions than in the XRD study. To investigate the globular core domains, we performed two-dimensional (2D) NCA and NCO, and dipolar assisted rotational resonance (DARR)[38] experiments for the human telomeric NCP containing $^{13}C,^{15}N$-labeled hH3, or hH4. As shown in Fig. 4, the SSNMR spectra of the human telomeric NCPs overlay well with those in the Widom 601 NCP, implying that the hH3 and hH4 adopt the same structural folds in the two nucleosomes. Small chemical shift perturbations are observed for hH4 A38, V43–S47, K79–T80, and R95–T96 in the C-terminus and hH3 Y54, T80, and I130, suggesting minor local conformational changes. Surprisingly, significant differences in peak intensities are observed, and a number of cross-peaks are not detected (Fig. 4a, b, d, e), indicating enhanced flexibility in the telomeric NCP. The quality of the reconstituted telomeric NCP was also verified on 6% native PAGE and 18% SDS–PAGE, ruling out the possibility that the absence of peaks is due to sample heterogeneity. To compare the site-specific dynamics of hH3 and hH4 in the telomeric NCP with that observed in the Widom 601 NCP,

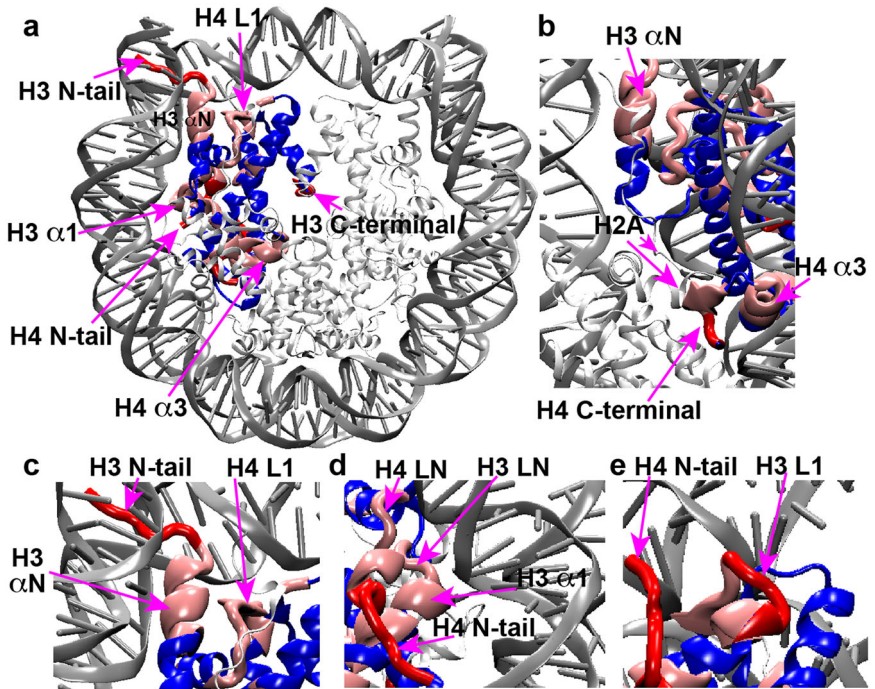

**Fig. 3 hH3 and hH4 residues in the NCP exhibiting pronounced microsecond–millisecond mobility form dynamic networks. a** Dynamic networks composed of H3 and H4 residues exhibiting pronounced μs–ms motions. Dynamics is highlighted on one copy of H3 and H4. Regions in red and pink are residues exhibiting negligible and relatively low CANCO peak intensities, respectively, that correspond to relatively larger-scale collective μs–ms motions. The rest of H3 and H4 are colored in blue. The rest histones and DNA are shown as light and dark gray ribbons with smaller thickness, respectively. **b–e** A zoomed-in view of these dynamic networks. Only one copy of each histones are displayed in **b–e**.

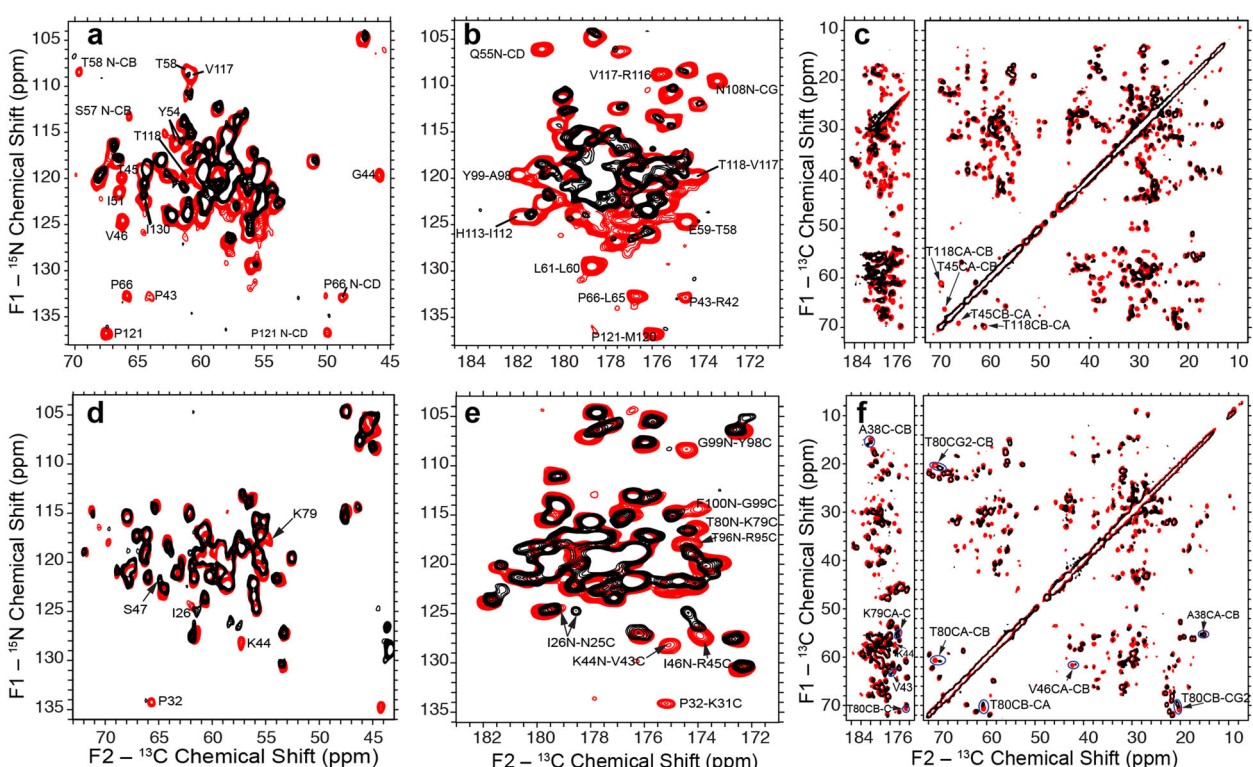

**Fig. 4 hH3 and hH4 in the telomeric NCP fold into the same structures as in the Widom 601 NCP with a few minor local conformational differences. a** Overlay of the 2D NCA, **b** NCO, and **c** $^{13}$C–$^{13}$C DARR SSNMR spectra of the Widom 601 NCP (red) and telomeric NCP (black) containing uniformly $^{13}$C, $^{15}$N-labeled hH3. **d** Overlay of the 2D NCA, **e** NCO, and **f** $^{13}$C–$^{13}$C DARR SSNMR spectra of the Widom 601 NCP (red) and telomeric NCP (black) containing uniformly $^{13}$C, $^{15}$N-labeled hH4. DARR spectra were collected with 20 ms mixing time.

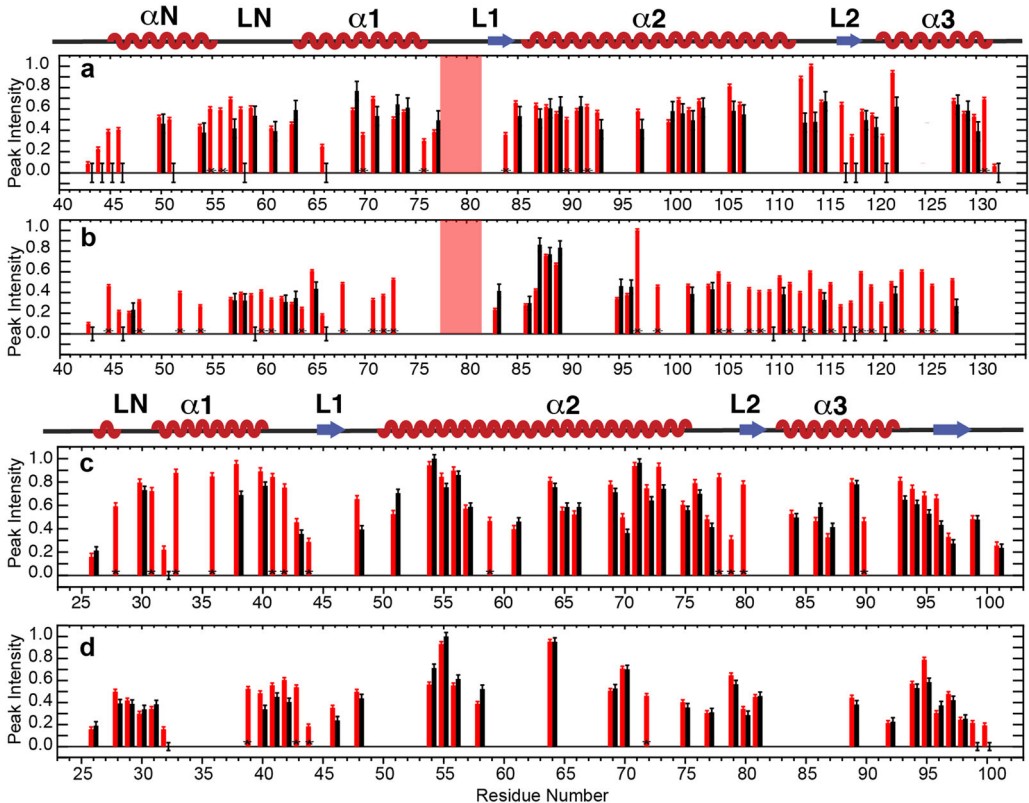

**Fig. 5 Several regions in the globular domains of hH3 and hH4 in the telomeric NCP possess relative enhanced microsecond–millisecond dynamics in comparison with the Widom 601 NCP. a** Normalized NCA and, **b** NCO peak heights of the Widom 601 NCP (red) and the telomeric NCP (black) containing $^{13}$C, $^{15}$N-labeled hH3. **c** Normalized NCA and, **d** NCO peak heights of the Widom 601 NCP (red) and the telomeric NCP (black) containing $^{13}$C, $^{15}$N-labeled hH4. The peak intensities were normalized to the highest peak intensity in the corresponding dataset. Error bars were derived from the RMSD values of spectral noise. Only residues in the hH3 P43–G132 region and the hH4 I26–G101 region and nonoverlapping in the Widom 601 NCP are shown. Red boxes in both graphs highlight residues F78–D81 of hH3 that are absent in both of the spectra of Widom 601 and telomeric NCP. Peaks that are absent in telomeric NCP are shown with zero intensity with error bars. Asterisks marks peaks that are resolved in Widom 601 NCP, but not in telomeric NCP (due to small chemical shift perturbations and peak broadening). The residues that have peak intensity differences exceeding error bars are considered as having different dynamical properties.

we plot the nonoverlapping NCA and NCO cross-peak intensities of hH3 and hH4 as shown in Fig. 5. Figure 6 shows the clusters of residues exhibiting lower relative NCA and NCO intensities, which correspond to larger-scale dynamics relative to other regions in the telomeric NCP in comparison with the Widom 601 NCP. The majority of residues with enhanced dynamics are found in the four networks (Fig. 6a–d) with the rest in spatial proximity to them (Fig. 6a, e). Previous SAXS and biochemical assay studies suggested that the telomeric NCP exhibits overall higher mobility in comparison with other canonical NCPs[22]. In addition, as discussed in the below section, our SSNMR data illustrates that the hH3 and hH4 N-terminal tails possess higher degree of flexibility in the telomeric NCP compared with the Widom 601 NCP. In summary, replacing the Widom 601 DNA with the telomeric DNA results in enhanced mobility in the identified dynamic networks and nearby motifs, leading to an overall more dynamic structure of the NCP formed by the unique repetitive G-rich telomeric DNA, which lacks positioning information. Thus, this provides another strong indication that the DNA in the nucleosomes can couple with the residues in those dynamic networks.

**hH3 and hH4 tails in Widom 601 and human telomeric NCP.** In the cell nucleus, histone N-terminal tails must remain high degree of mobility even in the condensed gene regions, such as

heterochromatin, to have the conformational flexibility to enable the interaction with various factors for gene regulation, which is indeed demonstrated by several studies. For example, the hH3 tails were shown to possess distinct conformations and are highly dynamic within NCP in solution[39], and the histones hH3 and hH4 N-terminal tails were previously demonstrated to exhibit significant mobility in the condensed nucleosome arrays[40]. Here, we first explore whether the hH3 and hH4 N-terminal tails remain mobile in the NCP of the most compacted state. We then assess the molecular differences at aa detailed level, introduced within the histone tails by altering the DNA sequence in the nucleosomes. As discussed in the above sections and the previous studies[23,31], the R42–G132 of hH3 and the I26–G101 of hH4 were observed in the dipolar-based SSNMR experiments in well-hydrated and highly compacted Widom 601 NCP (50–65% water content by weight corresponding to 500–350 mg/mL nucleosomes). In contrast, residues in or close to the N-terminal tails are absent in those spectra due to the significantly higher mobility. To probe the conformation and dynamics of the histone N-terminal tails in the tightly compacted NCP, $^{1}$H–$^{13}$C/$^{15}$N correlation SSNMR spectra were collected using 2D refocused J-based INEPT pulse schemes, which can detect components with significant mobility in protein samples. The $^{1}$H–$^{13}$C and $^{1}$H–$^{15}$N spectra are shown in Fig. 7b, d for hH3 and hH4, respectively, in the NCP reconstituted from the 145 bp Widom 601 DNA and precipitated with 20 mM Mg$^{2+}$, which is known to form ordered columnar

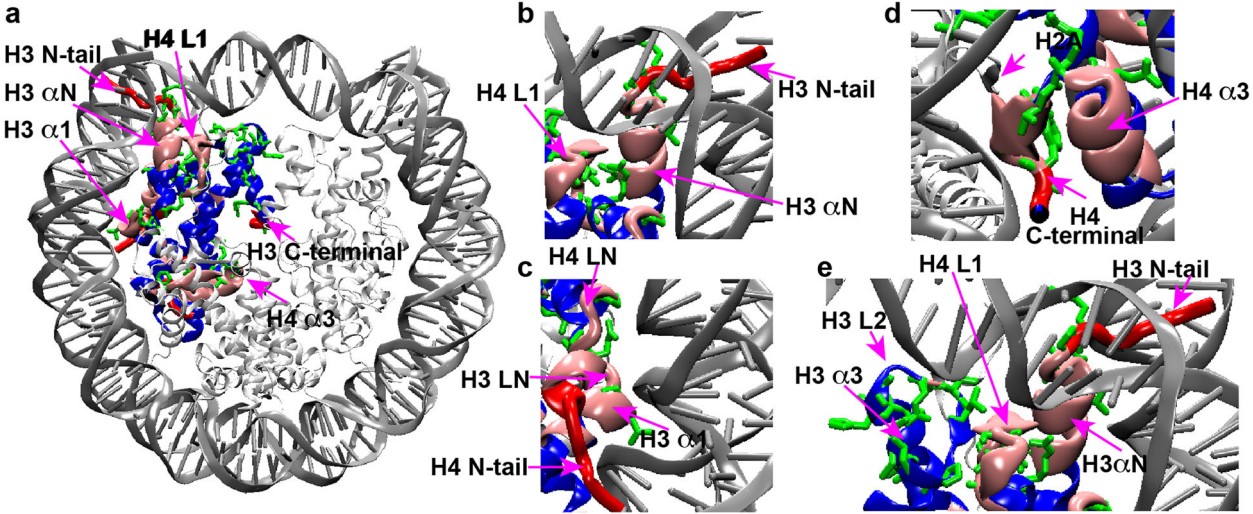

**Fig. 6 hH3 and hH4 residues exhibiting enhanced microsecond–millisecond dynamics in the telomeric NCP compared with the Widom 601 NCP are primarily reside at the identified dynamic networks and a neighboring region. a** H3 and H4 residues exhibiting lower relative NCA and NCO intensities that correspond to enhanced microsecond–millisecond dynamics in the telomeric NCP, as compared with the Widom 601 are shown in a chemical bond representation (green) in the NCP structure. The dynamic networks formed by H3 and H4 flexible regions are mapped onto the structure as shown in Fig. 4. **b–e** A zoomed-in view of the corresponding regions. Only one copy of each histone are displayed in **b–e**.

hexagonal stacking NCP assemblies[41]. The peaks are assigned by comparing with the spectra and assignments obtained from multidimensional liquid-state NMR experiments of the NCP dissolved in solution without $Mg^{2+}$. As displayed in Fig. 7b, d, A1–V35 of hH3 and S1–V21 of hH4 are observed in the precipitated Widom 601 NCP. This suggests that the histone N-terminal tails stay mobile in the most condensed nucleosome state, which enables the participation in the regulation of heterochromatin. The well-overlaid $^1H$–$^{13}C/^{15}N$ correlation SSNMR spectra and liquid-state HSQC spectra (Supplementary Fig. 1) indicate that histone N-terminal tails in the highly compacted NCP form the same random coil conformations, as in solution based on the observed chemical shift values. In addition, due to the intermediate mobility relative to the flexible tails and rigid globular domains, some degree of conformational heterogeneity is expected for K36–Y41 of hH3 and L22–N25 of hH4 as evidenced by their absence in these $^1H$–$^{13}C/^{15}N$ correlation spectra and in our previously reported SSNMR study detecting the rigid core components[23,31]. It was previously shown that hH3 N-terminal tails robustly form intermolecular contacts with the DNA in NCP[39,42]. Furthermore, the hH4 N-terminal tail can interact with DNA in addition with the H2A–H2B acidic patch of the neighboring nucleosomes[43]. Herein, we examine whether varying the DNA sequence will lead to the conformational and/or dynamical changes of hH3 and hH4 tails in the condensed NCP. The $^1H$–$^{13}C$ and $^1H$–$^{15}N$ spectra of the precipitated telomeric NCP are displayed in Fig. 7c, e and are compared with those of the Widom 601 NCP. The majority of the peaks overlay well, suggesting that the hH3 and hH4 N-terminal tail residues exhibit high mobility in the telomeric NCP as well. In the $^1H$–$^{13}C$ spectrum of hH3, a few peaks, including T22, A24, A25, and S28 CA–HA show perturbations in the telomeric NCP. In addition, those residues, as well as Q5 and R8 possess $^1H$–$^{15}N$ chemical shift differences between the two NCPs. These observations suggest that altering the DNA sequence in the NCP changes the chemical environment for the hH3 N-terminal tails in the nucleosomes, illustrating that the hH3 conformations tightly correlate to the DNA sequences, likely due to the direct interaction between the two. Furthermore, a few extra CA–HA peaks are present in the

$^1H$–$^{13}C$ spectrum of hH3 in the telomeric NCP. These peaks possibly belong to the residues between K36–R42. This is also evidenced by the extra Pro CD–HD peak, which can only be assigned to P38 since this is the only extra Pro within the highly flexible tail region or close to the globular domain. Thus, the region that exhibits significant mobility in hH3 N-terminal tail extends further toward the globular domain in the telomeric NCP. Similarly, the incorporation of telomeric DNA introduces differences to hH4 in the NCP; however, fewer residues are affected in comparison with the case of hH3. The hH4 tail forms a contact with the acidic patch of the H2A–H2B dimer of the neighboring NCP through K16–N25 (ref. [3]) and the S1–A15 is hypothesized to be involved in the interaction with DNA[43]. We observed small chemical shift differences for S1 CA–HA and CB–HB of hH4 in telomeric NCP compared with the Widom 601 NCP (Fig. 7e). In addition, a Gly $^1H$–$^{15}N$ peak at 8.82–109.7 p.p. m. is observed in the telomeric NCP, but not in the Widom 601 NCP (Fig. 7e). As there is no Gly in the L22–N25 stretch, we presume that this extra Gly peak belongs to one of the Gly observed in the Widom 601 NCP that possesses small chemical shift perturbations in the telomeric NCP, although we cannot pinpoint the exact residue due to peak overlapping. Overall, replacing the Widom 601 DNA with the telomeric DNA results in some conformational changes within the hH4 N-terminal tails; however, the affected residues are not as many as in the hH3 tails. This agrees with the participation of direct interaction with DNA. Interestingly, two distinct S1 CB–HB peaks are observed for hH4 in the telomeric NCP, suggesting two local conformations. Furthermore, the L10 CA–HA and CB–HB peaks are absent in the Widom 601 NCP, but present in the telomeric NCP, illustrating that the hH4 tails exhibit higher degree of flexibility in the later sample. Overall, we observed the first 35 residues of hH3 and the first 21 residues of hH4 in the Widom 601 NCP, confirming that these regions of the histone N-terminal tails possess significant mobility in the nucleosomes in their most compacted states. In the telomeric NCP, these tails become more dynamic in comparison with the Widom 601 NCP. It is worth noting that although hH3 and hH4 likely interact with DNA in these condensed NCPs, they possess a high degree of conformational

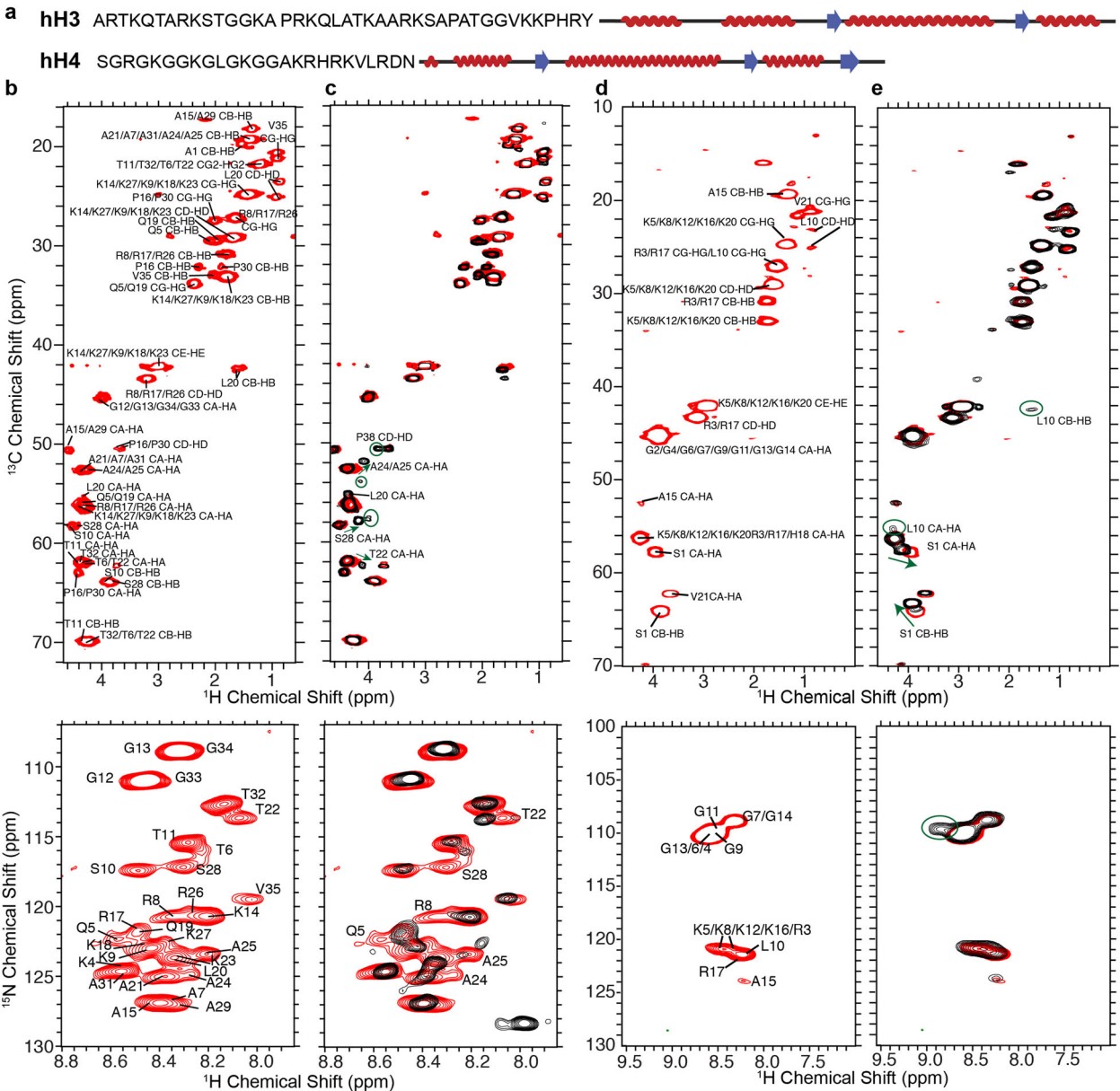

**Fig. 7 hH3 and hH4 tails are more dynamic in the telomeric NCP in comparison with the Widom 601 NCP. a** Schematic representations of hH3 and hH4 tail sequences and secondary structures. **b** $^1$H–$^{13}$C (upper panel) and $^1$H–$^{15}$N (lower panel) correlation SSNMR spectra of the Widom 601 NCP containing uniformly $^{13}$C,$^{15}$N-labeled hH3. **c** The overlaid $^1$H–$^{13}$C (upper panel) and $^1$H–$^{15}$N (lower panel) correlation SSNMR spectra of the Widom 601 NCP (red) and telomeric NCP (black) containing uniformly $^{13}$C,$^{15}$N-labeled hH3. **d** The $^1$H–$^{13}$C (upper panel) and $^1$H–$^{15}$N (lower panel) correlation SSNMR spectra of the Widom 601 NCP containing uniformly $^{13}$C,$^{15}$N-labeled hH4. **e** The overlaid $^1$H–$^{13}$C (upper panel) and $^1$H–$^{15}$N (lower panel) correlation SSNMR spectra of the Widom 601 NCP (red) and telomeric NCP (black) containing uniformly $^{13}$C,$^{15}$N-labeled hH4. These spectra were obtained with 2D refocused *J*-based INEPT pulse schemes. In **c, e**, arrows point the peaks shifting between the Widom 601 NCP and the telomeric NCP, circles highlight peaks observed in the telomeric NCP, but not in the Widom 601 NCP.

flexibility as distinct peaks are observed for the residues in these experiments detecting dynamic components.

## Discussion

The plasticity of the nucleosome core has been demonstrated to associate with DNA translocation and contributes to the gene regulation by chromatin factors[15–18]. The current study determined the dynamics of the hH3 and hH4 globular domains at two different timescales. The fact that no significant variation was observed for the ns–µs mobility across the hH3 in the Widom 601 NCP further conforms that the nucleosome core is compact. On the other hand, the µs–ms dynamics exhibits considerable

variation among different regions of the histones in the NCP. The observations of relatively lower CANCO peak intensities that are localized in distinct regions spanning both hH3 and hH4 histones clearly show the clustering of more dynamic aa residues. This strongly suggests that the µs–ms dynamics are correlated although the motional timescales, and amplitudes cannot be quantified by the present experiments. The presence of such networks immediately opens up the possibility of coupling of the nucleosome core with DNA and histone tails. This furthermore suggests that perturbations induced by epigenetic modifications in the core might be transmitted to DNA, thus, potentially altering its accessibility and function in an allosteric manner.

Compared with the Widom 601 NCP, the histone structures in the telomeric NCP are highly conserved; however, a significant increase in dynamics is observed for residues in the observed dynamic networks and neighboring motifs, further conforming coupling between the nucleosome core and DNA. It was recently proposed that nucleosome can be treated as an allosteric scaffold, and that binding/modifications can regulate gene activity by changing DNA mobility and accessibility at distal sites[14]. In addition, it has been suggested from the studies of chromatin-remodeling systems that the modulations of gene activities are achieved in an allosteric manner. Functional allosteric behavior has been increasingly extended to include long-range correlations of structural and/or dynamical changes within a subdomain of a system[44,45]. The allosteric response can be induced by perturbations of the internal correlated motions without structural changes that often occur through the pre-existing dynamic networks[45–48]. The currently revealed distinct dynamic clusters in nucleosomes extend from the center of the nucleosome core to DNA, which may serve as a pathway to propagate long-range changes to achieve DNA regulation by modifications in an allosteric fashion. This study points to the possibility of a novel mechanism of regulating gene activity by PTMs in the nucleosome core and the incorporation of variants. Experiments extending this study to the histones hH2A and hH2B, as well as the investigation of the effects of selective mutations at critical positions in the networks will be needed to shed further light on the details of the present findings. Additional quantitative characterization on the nature of these dynamic features can also be obtained to further understand the correlation of the motions and the functional relevance.

## Methods

**Preparation of nucleosome core particles**. Two Widom 601 NCP samples were reconstituted from 145 bp Widom 601 DNA and human histones containing uniformly $^{13}C$, $^{15}N$-labeled hH3 or $^{13}C$, $^{15}N$-labeled hH4. The human histones (sequences were harbored in pET-3a plasmids) were overexpressed using *Escherichia coli* BL 21 (DE3) pLys S grown in the 2× YT media. When the $OD_{600}$ reached 0.5, the overexpression was induced with IPTG at a concentration of 0.4 mM. The crude histones (in 7 M guanidium HCl, 20 mM sodium acetate at pH 5.2, and 10 mM DTT) were purified by gel filtration using HiPrep 26/60 Sephacryl S-200 HR column (GE Healthcare) under the denaturing condition (7 M urea, 20 mM sodium acetate at pH 5.2, 1 M NaCl, 5 mM BME, and 1 mM EDTA). The cation exchange chromatography with a Resource-S column (GE Healthcare) was used to eliminate DNA contamination. The uniformly $^{13}C$, $^{15}N$-labeled hH3 and $^{13}C$, $^{15}N$-labeled hH4 were overexpressed using *E. coli* BL 21 (DE3) pLys S. The cells were grown in the 2× YT media, and were pelleted down and resuspended in the M9 minimal media (0.2% $^{13}C$ glucose and 0.1% $^{15}N$ ammonium chloride, micronutrients, trace metals, and BME-vitamins) when the $OD_{600}$ reached 0.5. To induce the overexpression, 0.4 mM IPTG was added in the cell culture at $OD_{600}$ of 0.8. The plasmid PUC19 containing eight units of the Widom 601 DNA flanked by restriction sites for EcoRV was amplified in *E. coli* DH5α. Alkaline lysis method, RNAse treatment, phenol extraction, and precipitation by polyethylene glycol (PEG) 6000 were performed to extract the plasmids. Subsequently, the plasmids were digested using EcoRV-HF restriction enzyme (NEB). The Widom 601 DNA was obtained by PEG 6000-mediated fractionation and the subsequent extraction, using a mixture of chloroform and isoamyl alcohol (24:1). To prepare the HO, the mixture of the four human histones at the equimolar ratio was dialyzed in the refolding buffer containing 2 M NaCl, 1 mM EDTA, 5 mM beta-mercaptoethanol in 10 mM Tris·HCl at pH 7.5 at 4 °C. The pure HO sample was obtained by gel filtration with a HiLoad 16/600 Superdex 200 pg column (GE Healthcare). The NCP samples were prepared using the salt gradient dialysis method[23,49]. The quality of NCPs was assessed by evaluating the DNA and HO on the 18% SDS–PAGE and 6% PAGE, respectively.

Two telomeric NCP samples were reconstituted from 145 bp telomeric DNA (ATC-(TTAGGG)$_{23}$TGAT) and human histones, of which either hH3 or hH4 were $^{13}C$, $^{15}N$ labeled. The plasmids harboring one repeat of 145 bp telomeric DNA containing 23 telomeric repeats flanked by EcoRV and AvaI restriction sites were purchased from Bio Basic Asia Pacific Pte Ltd. DNA fragments containing eight repeats of the 145 bp telomeric DNA was prepared by Ava1 mediation self-ligation[50]. The DNA fragments were purified and cloned into a pUC57 plasmid and transformed into Sure2 *E. coli* (Agilent Technologies Singapore Pte.Ltd)[50]. To prepare the large-scale of plasmid DNA, cells were grown in TB medium (24 g/L yeast extract, 12 g/L tryptone, 4 mL/L glycerol, 17 mM $KH_2PO_4$, and 72 mM

$K_2HPO_4$) at 30 °C for 18 h. The plasmid was digested at a concentration of 2 mg/mL by overnight incubation at 37 °C using 75 units of enzyme per 1 mg plasmid. Subsequently, monomeric telomeric DNA was extracted from the vector DNA by employing PEG 6000-mediated fractionation with 10 mM $MgCl_2$. The blunt-ended DNA fragments were precipitated from the supernatant with ethanol (3× by volume) and then were resuspended in TE buffer (10 mM Tris pH 7.5, 0.1 mM EDTA). Traces of vector and recombination products were removed by ion exchange using a Mono Q5/50 GL column (GE Healthcare Pte Ltd Singapore). Briefly, the sample was injected into the column at 0.5 mL/min and then was washed with buffer A (ten column volumes) consisting of 300 mM LiCl, 20 mM Tris pH 7.5, and 1 mM EDTA to remove the remaining PEG 6000. Subsequently, a total of 55 column volumes of buffer B (750 mM LiCl, 20 mM Tris pH 7.5, and 1 mM EDTA) were loaded with a linear gradient of 40–65%. The eluted DNA fractions were combined and buffer exchanged with TE (10 mM Tris pH 7.5 and 0.1 mM EDTA) using a 10 kDa MWCO concentrator (Merck Pte. Ltd, Singapore). The purity of the obtained telomeric DNA was verified with 10% PAGE and 1.5% agarose gel electrophoresis.

The telomeric NCPs were reconstituted by the salt gradient dialysis method. A mixture of 20 mM Tris (pH 7.5), 2 M LiCl, 1 mM EDTA, 1 mM DTT, 4.8–7.2 μM HO, and 6 μM DNA were added in a dialysis bag (10 kDa MWCO). The mixture was first equilibrated against high-salt buffer (20 mM Tris pH 7.5, 2 M LiCl, 1 mM EDTA, and 1 mM DTT) for 30 min at room temperature. A low-salt buffer (20 mM Tris pH 7.5, 1 mM EDTA, and 1 mM DTT) was slowly pumped into the dialysis buffer; meanwhile, an equal amount of the mixed buffer was withdrawn from the dialysis buffer. Overall, 1.5 L low-salt buffer was continuously pumped into 0.6 L high-salt buffer over 18 h, and then the dialysis sample was incubated in the low-salt buffer for another 4 h. The quality of the reconstituted telomeric NCP was verified with 6% PAGE and 18% SDS–PAGE.

The NCP pellet was prepared by precipitating the NCP solution with $Mg^{2+}$ (ref. [41]). The pellets were transferred into SSNMR rotors (Bruker) using ultracentrifugation at $100,000 \times g$ for 1–3 h with rotor packing devices (Giotto Biotech).

**Liquid-state NMR experiments**. Liquid-state experiments were conducted on an 18.8 T Bruker Advance III HD spectrometer equipped with a 5 mm QCI H/P/C/N CryoProbe. The NCP samples containing isotopically labeled hH3 or hH4 were dissolved in 20 mM Tris (pH 7.5), 1 mM EDTA, 1 mM DTT, and 0.04% sodium azide 92/8 $H_2O/D_2O$ solution. Multidimensional experiments were performed for the Widom 601 NCPs with concentrations of 0.15–0.3 mM (pH 7.5) at 25 °C.

**Solid-state NMR experiments**. SSNMR experiments were performed on an 18.8 T Bruker Advance III HD spectrometer equipped with either a 3.2 mm HCN EFree MAS probe (for Widom 601 NCP containing isotopically labeled hH3 or hH4) or a 1.9 mm HCN MAS probe (for telomeric NCPs containing isotopically labeled hH3 or hH4). The actual sample temperature was controlled at 11–13 °C (calibrated externally with ethylene-glycol[51]). $^{13}C$ chemical shifts were referenced with adamantine using DSS scale (downfield signal at 40.48 p.p.m.) and $^{15}N$ chemical shift was indirectly calculated[52]. The typical 90° pulse lengths of $^1H$, $^{13}C$, and $^{15}N$ were 2.5, 3.6, and 5.0 μs, respectively, with the 3.2 mm probe, and were 2.2, 3.2, and 4.15 μs, respectively, with the 1.9 mm probe. 2D CC DARR[38], NCA, NCO, and 3D CANCO were collected and the detailed experimental parameters are summarized in Supplementary Table 1.

For the Widom 601 NCP containing $^{13}C$, $^{15}N$-labeled hH3, the 3D DIPSHIFT[32] experiments were conducted at 15.151 kHz with the 3.2 mm probe, and the NCA transfer was achieved by 4 ms SPECIFIC-CP, and the $^1H–^{15}N$ and $^1H–^{13}C$ dipolar coupling were reintroduced by applying R12$_1$ (ref. [4]) symmetry sequences[53,54] on $^1H$ channel in a constant-time manner. Data were processed using nmrPipe[55] and analyzed with Sparky (T. D. Goddard and D. G. Kneller, University of California, San Francisco). The $^1H–^{15}N$ and $^1H–^{13}C$ dipolar coupling line shapes were extracted from the 3D DIPSHIFT experiment, and are fitted using SIMPSON[56]. The line shapes in the frequency domain were obtained via Fourier transform of the experimental dipolar coupling trajectories with zero-filling to 256 points. The regions of −2 to 2 kHz of the $^1H–^{15}N$ dipolar line shapes and −4 to 4 kHz of the $^1H–^{13}C$ dipolar line shapes were considered in the fitting. The error bars plotted in Fig. 1 were the calculated 95% confidence intervals.

**Statistics and reproducibility**. The 3D experiments used to extract dynamics information were performed eight to nine times and were co-added together. The error bars (calculated 95% confidence intervals) of the SSNMR dipolar line shape fitting and the root-mean-square deviation values of peak intensities are presented in the figures in the main text.

**Reporting summary**. Further information on research design is available in the Nature Research Reporting Summary linked to this article.

## Data availability

The data supporting the findings of this study are available within the paper and the Supplementary Information. All relevant data is readily available upon request from corresponding authors.

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

## Acknowledgements

We thank Dr. Nikolay Korolev for comments on the manuscript. This work was supported by the Singapore Ministry of Education Academic Research Fund (AcRF) Tier 2 (MOE2018-T2-1-112) and Tier 3 (MOE2012-T3-1-001). All SSNMR experiments were performed at the Nanyang Technological University (NTU) Center of High Field NMR

Spectroscopy and Imaging. We also acknowledge the NTU Institute of Structural Biology (NISB) for supporting this research.

## Author contributions

X.S. performed the NMR experiments and data analysis, and contributed to the sample preparation. C.P. prepared the samples and contributed to the NMR experiments and data analysis. A.S. contributed to the preparation of the telomeric NCP samples. X.S. and L.N. conceived the study and wrote the manuscript with input and comments from all the co-authors. K.P. contributed to conceiving the study. L.N. acquired funding for the project.

## Competing interests

The authors declare no competing interests.
