## [Peer Review File · Communications Biology]

Reviewers' comments:

Reviewer #1 (Remarks to the Author):

This paper describes two major results. First, solid-state NMR experiments are used to determine the site-specific dynamics of histone H3 within the nucleosome core. These data are combined with similar histone H4 data from the authors' previous publications to describe the internal dynamics of the H3/H4 tetramer within the nucleosome. The combined experiments reveal a network of residues on both H3 and H4 that experience enhanced μ s-ms motions, a dynamic network that links the histone proteins to DNA. Second, the authors compare the spectra and the dynamics of a "601" nucleosome, the golden standard for biophysical experiments, with the dynamics of a "telomeric" nucleosome which represents a more native DNA sequence. As expected, the telomeric nucleosome is more dynamic, with enhanced motions associated with the previously identified dynamic network residues. Unexpectedly, some small chemical shift changes and peak doubling are observed for the dynamic tails as well.

While there are many crystal and cryo-EM structures of the nucleosome, this study represents the first complete analysis of the internal dynamics of the H3/H4 tetramer within the context of a full nucleosome and with two different types of DNA. These dynamics are important as they could allow for buried regions of the nucleosome to be exposed for protein binding and may be exploited by other chromatin binding proteins such as transcription factors and remodelers. Therefore, the study should be of interest to the broader biological community and should be valuable to chromatin structural biologists and biophysicists. Furthermore, the methodology should be applicable to the analysis of the H2A/H2B dimer and in other binding studies that involve the nucleosome.

Overall, I found the study to be well written and the data support the authors' conclusions well. The NMR data are of high quality and the experiments are described in sufficient detail. I only have a couple of minor questions. Regarding the figures, what are the peak intensities in Fig. 2 and 5 normalized to? And how many scans or total experimental time were necessary to record spectra of sufficient quality for comparison?

Reviewer #2 (Remarks to the Author):

The manuscript of Shi et al. describes an NMR study of histone dynamics in the histones H3 and H4 in both 601 and telomeric nucleosomes. This is very exciting and important question in the field of chromatin biology. The main claim of the paper is that several clusters of increased local dynamics are identified in the H3-H4 tetramer and that dynamics are enhanced in nucleosomes containing TTAGGG-repeat DNA, corresponding to telomeric nucleosomes.

The paper extends their previous work on H4 dynamics in 601 nucleosomes by here adding the data for H3. Dynamics are assessed from DIPSHIFT dipolar coupling profiles to extract CA-HA and N-H order parameters on ns-us time-scale and from CACNCO relative peak intensities to extract a per-residue qualitative indication of us-ms dynamics. Small clusters of residues with decreased order parameters or peak intensities are identified in the data to pinpoint regions of increased dynamics in the nucleosome structure. Since a number of these regions cluster in space, this is interpreted as evidence that these regions form a dynamic network.

These observations are then contrasted with dynamics data obtained for H3 and H4 in nucleosomes reconstituted from TTAGGG-repeat DNA, corresponding to telomeric nucleosomes, here using relative peak intensities in NCA and NCO spectra as indicator of us-ms dynamics. For several residues lower relative peak intensities compared to 601 are obtained, indicating increased dynamics. These residues primarily cluster in the areas of increased dynamics in the 601 nucleosome. In addition, J-based spectra of the histone tails indicate that the H3 and H4 tails are

more dynamic in the telomeric nucleosome because a number of additional resonances are observed in the telomere sample.

Overall the manuscript reports interesting observations that help to shape the thinking in the field. Yet a number of issues need to be addressed to substantiate the claims:

First, on the presentation and interpretation of the NMR data:

- Fig. 1: error bars in fitted S2 values are missing. Without error bars identification of outliers is not meaningful. Please include error bars and indicate how these were derived, e.g. fit-error, error from Monte Carlo resampling, F-statistic based ...
- Fig. 1 and 2: how are the outliers that are subsequently plotted on the structure identified? Is there an objective criterion? Please add this information.
- Fig. 2 and 3: how are the outliers divided over two classes for the color structure plot? Is there an objective criterion? Please add this information.
- Fig. 5: It is hard to tell from this figure which residues are considered to have a significant difference in relative peak intensity between 601 and telomeric nucleosomes. Please add an indicator (color/symbol) and add information on what criterion is used to identify these residues
- Fig 2. (and 5): Peak intensities in CP-based NCA/CO experiments indeed correlate with us-ms dynamics. (It would by the way be nice to add a few sentences to the main text describing the basic principle of this correlation as it is the foundation for the main claim of the paper). Yet it is not so clear how strict this relationship is in practice: strict application would indicate that there are two rigid residues (111 and 114) and the overall majority is rather dynamic. So in other words if outliers above 0.5 relative intensity are not considered meaningful, why are outlier below 0.5 relative intensity meaningful?

Second, on the interpretation of dynamic networks:

- the clustering of residues with decreased peak intensities is taken as sign of a dynamic network. While not explicitly defined, this suggest these residues experience some correlated motion. It should be mentioned clearly that the observation of differential peak intensities can only be taken as a sign of comparatively more or less dynamics and that it does not give any information on the nature of motion, the amplitude, exact time scale, or whether these motions are correlated or not. As such the claim that the study identifies, reveals or demonstrated four dynamic networks is over-stated and should be toned down.

Finally, the manuscript contains some debatable literature attributions and omits some relevant citations:

- p.2, l.21: a reference to the work of Xiang & Paige et al is appropriate.
- p.2, l.26: an explicit mention and reference to the work of Kivetski-LeBlanc et al on nucleosome dynamics is appropriate.
- p. 6, l.7: the paper of Morrison et al. provides some experimental evidence for histone tail-DNA interactions, but they were not the first. The paper from Stutzer et al Mol Cell 2016 provides more clear experimental evidence, so a reference to their work is appropriate.
- p.6, l.9: the crystal structures reported in the paper of Wakamori et al. indeed demonstrate an interaction between the H4 tail and the H2A/H2B acidic patch, yet this interaction was already well established at that point from the original nucleosome crystal structure and the paper from Chodaparambil Nat Struc Mol Biol 2007, so a reference to their work is appropriate

Textual issues:

- instead of referring to peaks as "site-resolved" it is better use "non-overlapping", p.16, l. 9 and p.5 l. 8
- The manuscript still contains a number of spelling errors, e.g. "telomierc ", p. 6, l.12, "temlomeric" on p 8, l 31.

Reviewer #3 (Remarks to the Author):

This is an interesting and timely study of histone proteins in nucleosomes using advanced NMR methods. An especially interesting aspect of the study is the comparison of histone protein structure and flexibility for different nucleosome positioning DNA sequences, including Widom 601 DNA and telomeric DNA. The quality of the NMR data presented are high, the results presented are likely to be of interest to the biophysics and chromatin biology communities and the paper is generally well-written. My main criticism of the manuscript in its current form, which may not be easily addressable but does not overly diminish my enthusiasm for this work, is that the functional significance of the purported networks of dynamic residues is quite speculative and unclear at this stage. A few additional points for the authors to consider are:

- 1) the relative amounts (and ideally hydration levels) of the different samples investigated should be clearly stated. Is it possible that some of the differences in peak intensities in multidimensional spectra that are being interpreted as arising solely from dynamic differences stem from different sample amounts and/or hydration levels?
- 2) Likewise additional comments on the sample conformational heterogeneity would be useful. The authors note that the samples with telomeric DNA appear homogeneous on a gel, but this does not address the issue of potential local heterogeneity? In other words, is it possible that some of the peak intensity differences are due to static disorder as opposed to increased dynamics on the microsecond-millisecond timescale?
- 3) The above concerns could possibly be addressed by measuring at least some R_2 or $R_{1\rho}$ rates as opposed to only comparing peak intensities and assuming that any differences arise from increased dynamics.

Reviewer #1:

This paper describes two major results. First, solid-state NMR experiments are used to determine the site-specific dynamics of histone H3 within the nucleosome core. These data are combined with similar histone H4 data from the authors' previous publications to describe the internal dynamics of the H3/H4 tetramer within the nucleosome. The combined experiments reveal a network of residues on both H3 and H4 that experience enhanced μ s-ms motions, a dynamic network that links the histone proteins to DNA. Second, the authors compare the spectra and the dynamics of a "601" nucleosome, the golden standard for biophysical experiments, with the dynamics of a "telomeric" nucleosome which represents a more native DNA sequence. As expected, the telomeric nucleosome is more dynamic, with enhanced motions associated with the previously identified dynamic network residues. Unexpectedly, some small chemical shift changes and peak doubling are observed for the dynamic tails as well.

While there are many crystal and cryo-EM structures of the nucleosome, this study represents the first complete analysis of the internal dynamics of the H3/H4 tetramer within the context of a full nucleosome and with two different types of DNA. These dynamics are important as they could allow for buried regions of the nucleosome to be exposed for protein binding and may be exploited by other chromatin binding proteins such as transcription factors and remodelers. Therefore, the study should be of interest to the broader biological community and should be valuable to chromatin structural biologists and biophysicists. Furthermore, the methodology should be applicable to the analysis of the H2A/H2B dimer and in other binding studies that involve the nucleosome.

We thank Reviewer #1 for his/her enthusiastic response to our manuscript.

Overall, I found the study to be well written and the data support the authors' conclusions well. The NMR data are of high quality and the experiments are described in sufficient detail. I only have a couple of minor questions.

R1 Q1. Regarding the figures, what are the peak intensities in Fig. 2 and 5 normalized to?

The peak intensities in Fig. 2 and 5 were normalized to the highest peak intensity in the corresponding dataset. This description was added in the captions of Fig. 2 and 5 in the revised manuscript.

R1 Q2. And how many scans or total experimental time were necessary to record spectra of sufficient quality for comparison?

In Table 1 in the revised Supplementary Information, we added the number of scans of all experiments and recycle delays that determine the total experimental times.

Reviewer #2:

The manuscript of Shi et al. describes an NMR study of histone dynamics in the histones H3 and H4 in both 601 and telomeric nucleosomes. This is very exciting and important question in the field of chromatin biology. The main claim of the paper is that several clusters of increased local dynamics are identified in the H3-H4 tetramer and that dynamics are enhanced in nucleosomes containing TTAGGG-repeat DNA, corresponding to telomeric nucleosomes.

The paper extends their previous work on H4 dynamics in 601 nucleosomes by here adding the data for H3. Dynamics are assessed from DIPSHIFT dipolar coupling profiles to extract CA-HA and N-H order parameters on ns-us time-scale and from CANCO relative peak intensities to extract a per-residue qualitative indication of us-ms dynamics. Small clusters of residues with decreased order parameters or peak intensities are identified in the data to pinpoint regions of increased dynamics in the nucleosome structure. Since a number of these regions cluster in space, this is interpreted as evidence that these regions form a dynamic network.

These observations are then contrasted with dynamics data obtained for H3 and H4 in nucleosomes reconstituted from TTAGGG-repeat DNA, corresponding to telomeric nucleosomes, here using relative peak intensities in NCA and NCO spectra as indicator of us-ms dynamics. For several residues lower relative peak intensities compared to 601 are obtained, indicating increased dynamics. These residues primarily cluster in the areas of increased dynamics in the 601 nucleosome. In addition, J-based spectra of the histone tails indicate that the H3 and H4 tails are more dynamic in the telomeric nucleosome because a number of additional resonances are observed in the telomere sample.

We thank Reviewer #1 for the enthusiastic response to our manuscript and the insightful comments to help us improving the manuscript.

Overall the manuscript reports interesting observations that help to shape the thinking in the field. Yet a number of issues need to be addressed to substantiate the claims. First, on the presentation and interpretation of the NMR data:

R2 Q1. Fig. 1: error bars in fitted S2 values are missing. Without error bars identification of outliers is not meaningful. Please include error bars and indicate how these were derived, e.g. fit-error, error from Monte Carlo resampling, F-statistic based.

We re-did the line shape simulations and fitting to extract the error bars, which was added in Fig.1 and the method of error analysis was described in Materials and Methods in the revised manuscript.

R2 Q2. Fig. 1 and 2: how are the outliers that are subsequently plotted on the structure identified? Is there an objective criterion? Please add this information.

As stated in the text in the last submitted manuscript, (shown in Fig. 1) the differences of the order parameters between different regions are rather small. Residues in the LN (the loop between α_N and α_1) and L1 exhibit slightly smaller order parameters compared to the rest of the globular domain, which is more clear after adding the error bars in the plots in the revised manuscript.

It was clearly stated in the last submitted manuscript that the CANCO peak intensities could be used to qualitatively track the μ s-ms dynamics. In Fig. 2, we classified the regions having residues with normalized peak intensities <0.5 as the more dynamic regions and those having residues with normalized peak intensities >0.5 as the more rigid regions. In order to be clear on the criterion used, we added the threshold (0.5) information in the Fig. 2 caption in the revised manuscript.

R2 Q3. Fig. 2 and 3: how are the outliers divided over two classes for the color structure plot? Is there an objective criterion? Please add this information.

This information was given in the Fig. 2 caption in the last submitted manuscript - “regions of H3 (one copy is shown) exhibiting negligible and relatively low peak intensities (<0.5) are in red and pink, respectively.”

R2 Q4. Fig. 5: It is hard to tell from this figure which residues are considered to have a significant difference in relative peak intensity between 601 and telomeric nucleosomes. Please add an indicator (color/symbol) and add information on what criterion is used to identify these residues.

The residues that have peak intensity differences exceeding error bars in the plots are considered as having different dynamical properties. To make this clear, we added this information explicitly in the Fig. 5 caption in the revised manuscript.

R2 Q5. Fig 2. (and 5): Peak intensities in CP-based NCA/CO experiments indeed correlate with μ s-ms dynamics. (It would by the way be nice to add a few sentences to the main text describing the basic principle of this correlation as it is the foundation for the main claim of the paper). Yet it is not so clear how strict this relationship is in practice: strict application would indicate that there are two rigid residues (111 and 114) and the overall majority is rather dynamic. So in other words if outliers above 0.5 relative intensity are not considered meaningful, why are outlier below 0.5 relative intensity meaningful?

Such information was provided in the last submitted manuscript - “The cross-peak intensities in dipolar-based heteronuclear correlation experiments such as CANCO and NCA are determined by the efficiencies of multiple dipolar heteronuclear transfers and can be used to qualitatively track the μ s-ms dynamics that interferes with relaxation rates ($T_{1\rho}$ and T_{2^*})^{23,30}”. This is a qualitative method to compare the dynamics of a residue relative to the others within a protein, we used 0.5 as the threshold to classify as relatively more dynamics and relatively more rigid.

R2 Q6. Second, on the interpretation of dynamic networks:

the clustering of residues with decreased peak intensities is taken as sign of a dynamic network. While not explicitly defined, this suggests these residues experience some correlated motion. It should be mentioned clearly that the observation of differential peak intensities can only be taken as a sign of comparatively more or less dynamics and that it does not give any information on the nature of motion, the amplitude, exact time scale, or whether these motions are correlated or not. As such the claim that the study identifies, reveals or demonstrated four dynamic networks is over-stated and should be toned down.

We had clearly stated that this method qualitatively track the μ s-ms dynamics. In the revised manuscript, we have changed the wording to be more cautious. In the Discussions section, we have stated these caveats as mentioned by the referee and also mentioned the need for further experiments to clarify the nature of these dynamic changes.

R2 Q7. Finally, the manuscript contains some debatable literature attributions and omits some relevant citations:

p.2, l.21: a reference to the work of Xiang & Paige et al is appropriate.

Added (ref 24).

R2 Q8. p.2, l.26: an explicit mention and reference to the work of Kivetski-LeBlanc et al on nucleosome dynamics is appropriate.

Added (ref 29).

R2 Q9. p. 6, l.7: the paper of Morrison et al. provides some experimental evidence for histone tail-DNA interactions, but they were not the first. The paper from Stutzer et al Mol Cell 2016 provides more clear experimental evidence, so a reference to their work is appropriate.

Added (ref 41).

R2 Q10. p.6, l.9: the crystal structures reported in the paper of Wakamori et al. indeed demonstrate an interaction between the H4 tail and the H2A/H2B acidic patch, yet this interaction was already well established at that point from the original nucleosome crystal structure and the paper from Chodaparambil Nat Struc Mol Biol 2007, so a reference to their work is appropriate.

The sentence discusses that the hH4 N-terminal tail can interact with DNA besides the H2A-H2B acidic patch, rather that hH4 N-terminal tail can interact with the H2A-H2B acidic patch. Therefore, those work were not cited here.

R2 Q11. Textual issues:

Instead of referring to peaks as “site-resolved” it is better use “non-overlapping”, p.16, l. 9 and p.5 l. 8

Corrected.

R2 Q12. The manuscript still contains a number of spelling errors, e.g. “telomiere “, p. 6, l.12, “temlomic” on p 8, l 31.

Corrected.

Reviewer #3

This is an interesting and timely study of histone proteins in nucleosomes using advanced NMR methods. An especially interesting aspect of the study is the comparison of histone protein structure and flexibility for different nucleosome positioning DNA sequences, including Widom 601 DNA and telomeric DNA. The quality of the NMR data presented are high, the results presented are likely to be of interest to the biophysics and chromatin biology communities and the paper is generally well-written. My main criticism of the manuscript in it's current form, which may not be easily addressable but does not overly diminish my enthusiasm for this work, is that the functional significance of the purported networks of dynamic residues is quite speculative and unclear at this stage.

We thank Reviewer #3 for the enthusiastic response to our manuscript and the insightful suggestions. This is the first work that shows the existence of the dynamic networks potentially enabling the communication between nucleosome core regions with histone tails and DNA. More extensive study on other histones in the nucleosomes as well as on the complexes of nucleosome-remodelers/factors are required to fully understand the functions of these dynamic networks, which is one of the directions of our future work.

A few additional points for the authors to consider are:

R1 Q1. The relative amounts (and ideally hydration levels) of the different samples investigated should be clearly stated. Is it possible that some of the differences in peak intensities in multidimensional spectra that are being interpreted as arising solely from dynamic differences stem from different sample amounts and/or hydration levels?

The sample amounts and hydration levels were added in the Supplementary Table 1 in the revised manuscript. The Widom ‘601’ NCP and telomeric NCP containing ^{13}C , ^{15}N labeled hH3 have a hydration level of 65% and 66%, respectively. The Widom ‘601’ NCP and telomeric NCP containing ^{13}C , ^{15}N labeled hH4 have a hydration level of 56% and 63%, respectively. These hydration levels are very close and correspond to well-hydrated pellets, which should not generate internal dynamics difference. In addition, the amount of samples does not induce structure and dynamics differences at the molecular level.

R1 Q2. Likewise additional comments on the sample conformational heterogeneity would be useful. The authors note that the samples with telomeric DNA appear homogeneous on a gel, but this does not address the issue of potential local heterogeneity? In other words, is it possible that some of the peak intensity differences

are due to static disorder as opposed to increased dynamics on the microsecond-millisecond timescale?

The fact that peaks in the DARR spectra overlay well indicates that there should not be any static disorder, otherwise, significantly broadened peaks would likely be observed in the DARR spectra as well as in the NCA and NCO spectra.

R1 Q3. The above concerns could possibly be addressed by measuring at least some R2 or R1rho rates as opposed to only comparing peak intensities and assuming that any differences arise from increased dynamics.

We agree that further extensive measurements such as relaxation experiments in combination of high/ultra-high MAS and sample deuteration will provide more information including quantified motional timescales and amplitudes, and more complete understanding of the dynamics and its function relevance. We added two sentences at the end of Discussion to point out the future directions towards understanding this. Those require substantial extra work, which we would explore in our following studies. The concerns raised here were fully explained in the answers for R3 Q1 and Q2.

REVIEWERS' COMMENTS:

Reviewer #1 (Remarks to the Author):

Overall, the authors have addressed the majority of the reviewers comments. The manuscript still needs to be checked carefully for typos. Also, the error bars in Fig. 2 and Fig. 5 should be based on signal-to-noise and not just the noise level, i.e. they shouldn't be the same for all peaks that come from the same spectrum.

Reviewer #2 (Remarks to the Author):

The authors have addressed most of my issues. Two points remain:

- the criterion for color coding in Fig. 1 is not stated
- the criterion for color coding in Fig. 2 does not match the color coding in the picture, i.e. few residues in the a2 helix also have relative peak intensities < 0.5 . This is also true for Figure 3.

My point on the interpretation of the relative intensity difference (Q5/6) also remains in the current version, there is an intrinsic problem in translating a gradually changing observable (relative peak intensities) into a binary outcome (more or less dynamic). In other words, when coloring simply the peak intensities onto the structure it will be a lot less clear how the differences can be interpreted. I concede that I may advocate a conservative approach here and leave this as a final remark.

Reviewer #3 (Remarks to the Author):

The authors have been able to satisfactorily address the comments of all the reviewers that could readily be addressed without performing extensive additional experiments. I find the revised manuscript to be significantly improved and clearer.

Responses to reviewers' comments:

Reviewer #1:

R1 Q1. The manuscript still needs to be checked carefully for typos.

We corrected typos and grammar errors as highlighted in the revised manuscript (COMMSBIO-20-1752A_Revision_Highlighted.docx).

R1 Q2. The error bars in Fig. 2 and Fig. 5 should be based on signal-to-noise and not just the noise level, i.e. they shouldn't be the same for all peaks that come from the same spectrum.

The root-mean-squared noise of solid-state NMR spectrum is commonly accepted as the error values of peak intensities, especially for NCA, NCO and CANCO type experiments. The root-mean-squared noise has been used in many publications (e.g. Gemma Comellas, et. al., Journal of American Chemical Society, 2012, 134, 5090-5099; Jun-Xia Lu, et. al., Cell, 2013, 154, 1257-1268; Peng Xiao, et. al., Nature Communications, 2019, 10, 3867). Therefore, we did not change the error values in Fig. 2 and 5.

Reviewer #2:

R2 Q1. The criterion for color coding in Fig. 1 is not stated.

We added information in the Fig.1 caption to clearly state the criterion of color-coding in the figure. The changes were highlighted in the revised manuscript (COMMSBIO-20-1752A_Revision_Highlighted.docx).

R2 Q2. The criterion for color coding in Fig. 2 does not match the color coding in the picture, i.e. few residues in the a2 helix also have relative peak intensities < 0.5. This is also true for Figure 3.

Fig. 2, 3 and 6 were re-plotted to correct the corresponding information. Residues 70, 71, 93, 101, 104 and 117 in H3 were highlighted in pink in Fig. 2, 3 and 6, and, correspondingly, sites of DNA and other histone sites that were ≤ 10 Å away from these six residues were highlighted in green in Fig. 2.

Reviewer #3 (Remarks to the Author):

The authors have been able to satisfactorily address the comments of all the reviewers that could readily be addressed without performing extensive additional experiments. I find the revised manuscript to be significantly improved and clearer.